# Harnessing the Computation Redundancy in ViTs to Boost Adversarial Transferability

Jiani Liu[1]♣    Zhiyuan Wang[2]♣    [†]Zeliang Zhang[1]♣    Chao Huang[1]    Susan Liang[1]
Yunlong Tang[1]    Chenliang Xu[1]
[1]University of Rochester    [2]University of California, Santa Barbara
♣Equal contribution listed alphabetically    [†]Project lead (✉: hust0426@gmail.com)

## Abstract

Vision Transformers (ViTs) have demonstrated impressive performance across a range of applications, including many safety-critical tasks. Many previous studies have observed that adversarial examples crafted on ViTs exhibit higher transferability than those crafted on CNNs, indicating that ViTs contain structural characteristics favorable for transferable attacks. In this work, we take a further step to deeply investigate the role of computational redundancy brought by its unique characteristics in ViTs and its impact on adversarial transferability. Specifically, we identify two forms of redundancy, including the data-level and model-level, that can be harnessed to amplify attack effectiveness. Building on this insight, we design a suite of techniques, including attention sparsity manipulation, attention head permutation, clean token regularization, ghost MoE diversification, and learning to robustify before the attack. A dynamic online learning strategy is also proposed to fully leverage these operations to enhance the adversarial transferability. Extensive experiments on the ImageNet-1k dataset validate the effectiveness of our approach, showing that our methods significantly outperform existing baselines in both transferability and generality across diverse model architectures, including different variants of ViTs and mainstream Vision Large Language Models (VLLMs). Our code[1] is publicly released at `https://github.com/Trustworthy-AI-Group/TransferAttack` under the name **LL2S**.

## 1   Introduction

Vision Transformers (ViTs), empowered by self-attention, have achieved state-of-the-art performance in various domains, including face forgery detection [Zhuang et al., 2022], 3D semantic segmentation for autonomous driving [Ando et al., 2023], and disease progression monitoring [Mbakwe et al., 2023], many of which are safety-critical. While deep neural networks (DNNs) are known to be vulnerable to imperceptible adversarial perturbations [Szegedy et al., 2013], most existing work focuses on general-purpose attacks and defenses, often tailored to convolutional architectures [Zhu et al., 2024a, Wang and Farnia, 2023, Wang et al., 2020, 2021a, Carlini and Wagner, 2017]. However, ViTs differ fundamentally from CNNs in representation and structure [Naseer et al., 2021, Raghu et al., 2021], leaving a gap in understanding their unique vulnerabilities. Designing attack strategies that leverage the distinctive properties of ViTs is essential for both exposing their weaknesses and building more robust models for real-world deployment [Song et al., 2025a].

Although adversarial perturbations can be crafted using a white-box model (*i.e.*, a surrogate model), numerous studies have demonstrated their threat to black-box models (*i.e.*, victim models) due to a phenomenon known as adversarial transferability [Zhou et al., 2018, Li et al., 2020]. Compared to white-box attacks, black-box adversarial attacks, some of which leverage transferability, typically show lower performance but offer greater practicality in real-world applications [Huang et al., 2024,

---

[1]The development version is available at `https://github.com/JennnyL/RedunAttack-ViT`.

Wang et al., 2019]. Recent research has primarily focused on designing more efficient black-box adversarial attacks, including gradient-based [Dong et al., 2018, Wang et al., 2021b], model-based [Li et al., 2020, 2023], input transformation-based [Xie et al., 2021, Zhu et al., 2021], among others.

While both CNNs and ViTs can serve as surrogate models, existing studies [Wang et al., 2023a, Zhu et al., 2024a] have found that adversarial examples crafted on ViTs tend to transfer more effectively, whereas attacking ViTs using examples crafted on CNNs remains notably challenging. Unlike CNNs, ViTs incorporate a tokenization mechanism and a sequence of shape-invariant blocks. These unique characteristics, such as tokens, attention mechanisms, and the chain of blocks, have motivated numerous studies to design adversarial attacks specifically tailored for ViTs [Zhang et al., 2023, Zhu et al., 2024b, Ren et al., 2025a]. We argue that the success of these methods is closely related to the computational redundancy inherent in ViTs.

In this paper, we thoroughly investigate the relationship between computational redundancy and adversarial transferability. While prior studies have shown that reducing computation in ViTs does not significantly degrade performance, we instead aim to leverage this redundancy to enhance the adversarial transferability of crafted perturbations. Based on a detailed analysis of computational redundancy in ViTs, we propose a collection of efficient techniques that exploit this property to improve adversarial transferability. An overview of our proposed methods is shown in fig. 1.

Our contributions can be summarized as follows:

1. We analyze the computational redundancy in ViTs and demonstrate how it can be effectively leveraged to boost adversarial transferability.

2. We propose a suite of effective methods that exploit computational redundancy to enhance transferability, including amplifying attention sparsity, permuting attention weights, introducing clean tokens for regularization, diversifying the feed-forward network via ghost MoE, and learning to robustify before the attack to improve the theoretical bound of adversarial transferability.

3. We propose an online learning strategy that leverages the proposed operations to learn redundantization, thereby boosting adversarial transferability and generalization.

4. We conduct extensive experiments on the ImageNet-1k dataset to validate the effectiveness of our approach. The results show that our methods outperform existing baselines by a clear margin across various models, demonstrating both their superiority and generality.

## 2 Related work

**Adversarial Transferability.** The vulnerability of deep neural networks (DNNs) [Min et al., 2024, Song et al., 2025b] to adversarial perturbations, first revealed by Szegedy et al. [2013], has triggered extensive research on both attack and defense strategies. Adversarial attacks are broadly categorized based on the attacker's access to the model into white-box and black-box attacks.

*White-box attacks* assume full access to the model architecture and gradients. Canonical examples include FGSM [Goodfellow et al., 2015], DeepFool [Moosavi-Dezfooli et al., 2016], and the Carlini & Wagner (C&W) attack [Carlini and Wagner, 2017]. In contrast, *black-box attacks* operate without such access and include *score-based attacks* [Andriushchenko et al., 2020, Yatsura et al., 2021], *decision-based attacks* [Chen et al., 2020, Li et al., 2022, Wang et al., 2022b], and *transfer-based attacks* [Dong et al., 2018, Lin et al., 2020, Wang et al., 2021a]. Transfer-based attacks are particularly appealing due to their query-free nature and strong cross-model performance. Our work focuses on enhancing this category. Existing methods to improve adversarial transferability can be grouped into three categories:

**Gradient-based Strategies.** These methods refine the optimization path to stabilize and generalize perturbations. Momentum-based attacks such as MI-FGSM [Dong et al., 2018] and NI-FGSM [Lin et al., 2020] enhance convergence stability, while PI-FGSM [Gao et al., 2020] and VMI-FGSM [Wang and He, 2021] introduce spatial and variance smoothing. EMI-FGSM [Wang et al., 2021b] averages gradients over multiple directions, and GIMI-FGSM [Wang et al., 2022a] initializes momentum from pre-converged gradients to boost transferability.

**Input Transformation Techniques.** These approaches modify the input space to produce more robust adversarial examples. DIM [Xie et al., 2019] applies random resizing and padding, TIM [Dong et al., 2019] uses gradient smoothing over translated inputs, and SIM [Lin et al., 2020] aggregates

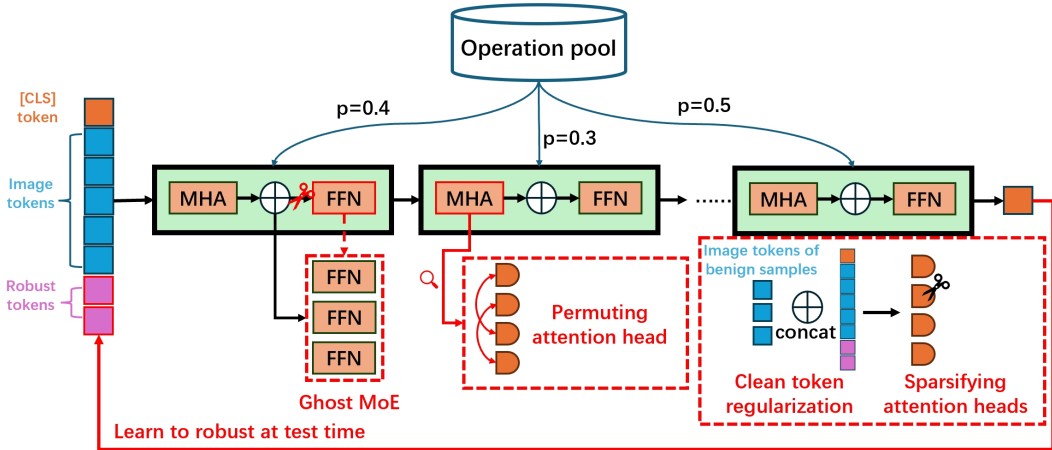

Figure 1: Overview of the proposed attack strategy integrated into Vision Transformers (ViTs). Our method adopts a policy gradient-based framework to selectively apply different operations from an operation pool to each transformer block. These operations include permuting attention heads, sparsifying them, clean token regularization, and activating auxiliary Ghost MoE branches to exploit the computational redundancy within ViTs. Robust tokens are learned at test time to further enhance adversarial transferability.

multi-scale gradients. Admix [Wang et al., 2021a] blends samples from different categories, while SSA [Long et al., 2022] perturbs images in the frequency domain.

**Model-Centric Approaches.** These methods diversify the surrogate model to reduce overfitting of adversarial perturbations. Liu et al. [2017] demonstrate that ensemble attacks increase transferability. Ghost networks [Li et al., 2020] simulate model variation using dropout, while stochastic weight averaging [Xiong et al., 2022] and high-learning-rate snapshots [Gubri et al., 2022] offer temporal diversity through model evolution.

## 3 From Computational Redundancy to Strong Adversarial Transferability

### 3.1 Preliminaries

**Vision transformer**. Given an input image $\mathbf{x} \in \mathbb{R}^{H \times W \times C}$, the Vision Transformer first splits the image into a sequence of $N$ patches, each of size $P \times P$. These patches are then flattened and projected into a latent embedding space using a learnable linear projection, *i.e.*, $\mathbf{z}^i = \mathbf{E} \cdot \text{Flatten}(\mathbf{x}_i), \quad i = 1, \ldots, N$, where $\mathbf{E} \in \mathbb{R}^{(P^2 \cdot C) \times D}$ is the patch embedding matrix and $D$ is the hidden dimension. A learnable class token $\mathbf{z}^{[\text{CLS}]}$ is prepended to the sequence, and positional embeddings $\mathbf{p}_i$ are added, *i.e.*, $\mathbf{z} = [\mathbf{z}^{[\text{CLS}]}, \mathbf{z}^1 + \mathbf{p}_1, \ldots, \mathbf{z}^N + \mathbf{p}_N]$. This token sequence is then passed through $L$ Transformer encoder layers. Each layer consists of a multi-head self-attention (MHA) mechanism and a feed-forward network (FFN), both wrapped with residual connections and layer normalization. The update for the $\ell$-th layer can be formulated as:

$$\mathbf{z}_\ell = \text{FFN}(\text{LN}(\text{MHA}(\text{LN}(\mathbf{z}_{\ell-1})) + \mathbf{z}_{\ell-1}) + \text{MHA}(\text{LN}(\mathbf{z}_{\ell-1})), \tag{1}$$

where $\text{LN}(\cdot)$ denotes layer normalization. MHA and FFN are defined as:

$$\text{MHA}(\mathbf{z}) = \text{softmax}\left(\frac{\mathbf{z}\mathbf{W}_Q(\mathbf{z}\mathbf{W}_K)^\top}{\sqrt{d_k}}\right)\mathbf{z}\mathbf{W}_V, \quad \text{FFN}(\mathbf{z}) = \text{GELU}(\mathbf{z}\mathbf{W}_1 + \mathbf{b}_1)\mathbf{W}_2 + \mathbf{b}_2, \tag{2}$$

where we denote $\mathbf{W}_Q, \mathbf{W}_K,$ and $\mathbf{W}_V$ as projection matrices for queries, keys, and values, $\mathbf{W}_1, \mathbf{W}_2$ as the weights of the two-layer feed-forward network, and $\mathbf{b}_1, \mathbf{b}_2$ as the biases. Last, only the $\mathbf{z}^{[\text{CLS}]}$ will be used for classification by a linear projection layer.

**Adversarial attacks**. The iterative generation of adversarial examples can be formulated as:

$$x_{i+1}^{\text{adv}} = \text{clip}_\epsilon\left(x_i^{\text{adv}} + \alpha \cdot \text{sign}(\delta_i)\right), \tag{3}$$

where $\text{clip}_\epsilon(\cdot)$ ensures that the perturbation remains within an $\ell_\infty$-norm ball of radius $\epsilon$ centered at the clean input $x$, and $\alpha$ is the step size. The update $\delta_i$ varies depending on the attack method.

## 3.2 Analysis of computational redundancy in ViTs

Computational redundancy in Vision Transformers (ViTs) exists at two main levels: data-level redundancy and model-level redundancy.

- *Data-level redundancy* has been extensively studied, particularly through token pruning techniques. Due to overlapping visual representations, many tokens carry similar information and can be selectively pruned at various stages of the ViT's processing pipeline. This allows for a more focused use of computation without affecting task performance.

- *Model-level redundancy* arises from over-parameterization and certain training strategies, such as neuron dropout in FFN modules and layer dropout across entire transformer blocks. Additionally, research has shown that not all attention heads in the MHA module contribute equally to performance. These findings suggest that parts of the model can be selectively deactivated or repurposed while maintaining accuracy.

These forms of redundancy present an opportunity to reallocate computational effort toward improving adversarial transferability, without altering the overall computational workload. We provide a verification study on these redundancies in appendix A.

To better understand the relationship between computational redundancy and adversarial transferability in ViTs, we conduct a series of experiments to validate our hypothesis. We randomly sample 1,000 images from the ImageNet-1K dataset as our evaluation set. Eight models are used as both surrogate and victim models, including (1) four Convolutional Neural Networks (CNNs): ResNet-50 [He et al., 2016], VGG-16 [Simonyan and Zisserman, 2015], MobileNetV2 [Sandler et al., 2018], Inception-v3 [Szegedy et al., 2016] and (2) four Transformer-based models: ViT [Dosovitskiy et al., 2020], PiT [Heo et al., 2021], Visformer [Liu et al., 2021] and Swin Transformer [Liu et al., 2021]. Adversarial examples are generated using ViT as the surrogate model, and their transferability is evaluated on the re-

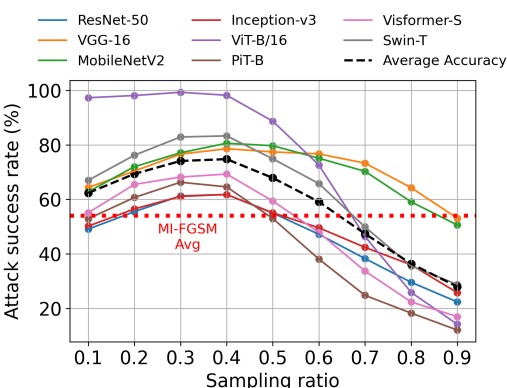

Figure 2: Study on the effectiveness of randomly dropping attention weights.

maining models. We benchmark the performance by MI-FGSM, and our proposed strategies are integrated into MI-FGSM. Following standard settings in prior work, set the maximum perturbation magnitude to $\epsilon = \frac{16}{255}$ with the momentum decay factor as 1 and use 10 attack steps for all methods.

## 3.3 Practical exploitation of redundancy for adversarial transferability

✂ *On the role of attention sparsity in adversarial transferability*. Prior studies have shown that Vision Transformers can maintain performance even when a subset of tokens is dropped at various layers. This implicitly alters the attention patterns and indicates that redundancy exists within the attention mechanism itself, which can potentially be repurposed for other objectives. These observations naturally raise the question: *can adversarial transferability be improved by actively manipulating attention sparsity?*

To exploit this, different from Ren et al. [2025a] which study the adversarial transferability by dropping attention blocks, we propose to diversify the attention maps by directly randomly dropping attention weights with a predefined ratio $r$. Specifically, we apply a binary mask $\mathbf{M}$ to the attention logits before the softmax operation in eq. (2), formulated as:

$$\text{MHA}(\mathbf{z}) = \text{softmax}\left(\left(\frac{\mathbf{z}\mathbf{W}_Q(\mathbf{z}\mathbf{W}_K)^\top}{\sqrt{d_k}}\right) \odot \mathbf{M}\right)\mathbf{z}\mathbf{W}_V, \tag{4}$$

where $\mathbf{M} \in \{0,1\}^{N \times N}$ is a randomly sampled binary mask with a drop ratio $r$, and $\odot$ denotes element-wise multiplication.

⚖️ **Results**. As shown in fig. 2, we vary the sampling ratio $r$ from 0.1 to 0.9 to investigate the transferability of adversarial samples generated on ViT across various target models. As the sampling ratio increases up to $0.4$, the white-box attack success rate remains consistently high, revealing that ViTs exhibit a notable degree of redundancy. Beyond this point, however, the white-box attack success rate begins to decline. In contrast, black-box attack success rates follow a rise-then-fall pattern, with peak transferability occurring at different sampling ratios depending on the target model. These results suggest that *moderate sparsification allows adversarial attacks to exploit attention redundancy in ViTs, enhancing perturbation transferability by focusing on fewer but more transferable features.*

However, excessive sparsification harms both white-box and black-box performance, revealing a trade-off between leveraging redundancy and preserving representational capacity. These findings align with those of Ren et al. [2025a], who drop attention blocks to study similar effects, whereas our approach directly controls sparsity at the element-wise level.

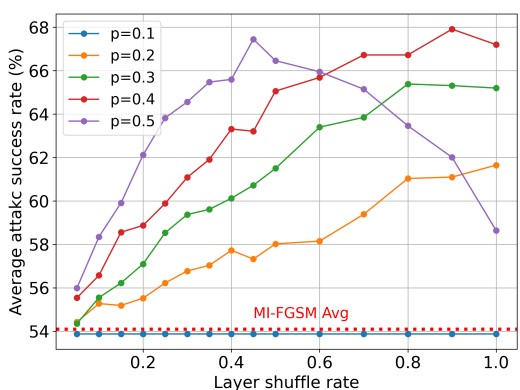

Figure 3: Study on the effectiveness of shuffling attention heads.

🏹 *Permuting attention heads to capture more generalizable features*. The multi-head attention mechanism enhances model capacity by allowing each attention head to focus on different subspaces of the input. However, in practice, many heads exhibit similar attention patterns, often attending to overlapping regions. This redundancy suggests the presence of invariant features that may be beneficial for adversarial transferability.

To better exploit this invariance, we propose to introduce randomness into the attention mechanism by permuting the attention weights among different heads. Specifically, during each attack iteration, we apply a random permutation to the attention heads (the group of QK layers). This encourages attention heads to explore diverse focus patterns while keeping the value projections unchanged. Rewriting the multi-head attention module in eq. (2), and incorporating the permutation operation, we obtain:

$$\text{MHA}(\mathbf{z}) = \text{Concat}\left(\text{softmax}\left(\pi\left(\frac{\mathbf{Q}_1\mathbf{K}_1^\top}{\sqrt{d_k}}, \frac{\mathbf{Q}_2\mathbf{K}_2^\top}{\sqrt{d_k}}, ..., \frac{\mathbf{Q}_H\mathbf{K}_H^\top}{\sqrt{d_k}}\right)\right)[\mathbf{V}_1, \mathbf{V}_2, ..., \mathbf{V}_H]^T\right), \quad (5)$$

where $\mathbf{Q}_h = \mathbf{z}\mathbf{W}_Q^h$, $\mathbf{K}_h = \mathbf{z}\mathbf{W}_K^h$, and $\mathbf{V}_h = \mathbf{z}\mathbf{W}_V^h$ denote the query, key, and value projections for the $h$-th head, respectively. $\pi(\cdot)$ represents a random permutation applied to the attention weights of each head, and is resampled at each iteration to promote diversity in attention patterns.

⚖️ **Results**. In our validation experiments, we study two factors, namely inter-layer and intra-layer randomness. Specifically, each layer has a probability $p$ of being selected for attention head shuffling, and a ratio $r$ of attention heads are randomly permuted within the selected layers. The results are shown in fig. 3. On the one hand, we observe that larger values of $p$ and $r$ generally lead to stronger adversarial transferability, while excessive disorder, such as $p = 0.5, r = 1.0$, can result in performance degradation. This experiment also validates our hypothesis that different attention heads learn similar visually robust regions of interest, which benefits the crafting of highly transferable adversarial examples. This suggests that *adversarial perturbations do not rely on specific heads, but rather exploit shared information across redundant attention heads to enable transferable attacks.*

🏹 *Introducing clean tokens to regularize adversarial representations*. Recall that only the $\mathbf{z}^{\texttt{[CLS]}}$ token is used for classification, while the remaining patch tokens are typically discarded after the final transformer layer. Prior work [Wang et al., 2023b] on attacking CNNs has shown that incorporating clean features into the forward pass can act as a strong regularization signal, significantly improving adversarial transferability. Inspired by this, we propose a strategy tailored to ViTs: at each transformer block, we append a small number of clean tokens from the benign samples alongside the adversarial ones. These clean tokens serve as a stabilizing anchor that helps regularize the evolving adversarial representations throughout the network, encouraging the model to preserve more transferable patterns.

⚖️ **Results**.  We scale the sampling ratio $r$ from 0.1 to 0.8, and present the results in fig. 4. As observed, incorporating clean tokens leads to a consistent improvement in attack success rate across most target models, particularly at moderate sampling ratios (e.g., $r = 0.3$ to $0.5$). This confirms the effectiveness of clean token injection as a form of regularization that strengthens adversarial transferability. However, we also note that excessive inclusion of clean tokens (e.g., $r > 0.4$) results in diminishing returns or slight degradation in performance. This suggests a trade-off between regularization and distortion of the adversarial signal. Overall, the results highlight that a small proportion of clean context is sufficient to guide the adversarial optimization towards transferable perturbations.

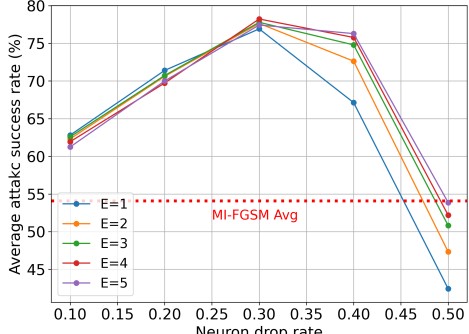

Figure 4: Study on the effectiveness of clean tokens in regularization.

🏹 *Diversifying the FFN via a ghost Mixture-of-Experts*.  Due to the use of dropout during training, neurons in the FFNs of exhibit a degree of functional redundancy and robustness, enabling alternative inference paths without significant performance degradation. To introduce additional computational redundancy that can be exploited for enhancing adversarial transferability, we propose a ghost Mixture-of-Experts (MoE) design. In this framework, each expert is instantiated by applying a distinct dropout mask to the original FFN, effectively creating multiple sparse subnetworks that share parameters but activate different neuron subsets. The inference process under this ghost MoE design is defined as:

$$\text{MoE}(\mathbf{z}) = \frac{1}{q} \sum_{e=1}^{q} \text{FFN}_{\theta_e}(\mathbf{z}), \qquad q \sim \mathcal{U}(1, E) \tag{6}$$

where $E$ denotes the maximum number of experts, and $\theta_e$ represents the $e$-th randomly perturbed configuration of the original FFN weights induced by a randomly sampled dropout mask. All experts share the same underlying parameters but differ in their active neurons due to stochastic masking.

⚖️ **Results**.  We scale the maximum number of experts $E$ from 1 to 5 and vary the neuron drop rate from 0.1 to 0.5. The results in fig. 5 show that increasing $E$ consistently improves performance, with the best results achieved at a drop rate of 0.3 for all configurations of the number of experts. Higher drop rates require more experts to offset the performance loss caused by representation collapse. This highlights the benefit of moderate sparsification and expert diversity in enhancing adversarial transferability. However, excessive dropout (e.g., $0.5$) leads to degradation, indicating a trade-off between diversity and feature preservation.

🏹 *Robustifying the ViT before attacking for better transferability*.  As observed by Bose et al. [2020], given an input $x$ with label $y$, the transferability of adversarial examples can be improved by attacking a robust model $f \in \mathcal{F}$. This can be formulated as the following adversarial game:

$$\min_{f \in \mathcal{F}} \max_{\delta} \mathcal{L}(f(x + \delta), y), \tag{7}$$

where $\delta$ denotes the adversarial perturbation and $\mathcal{L}$ is the classification loss. Motivated by this, a natural idea is to adversarially train a surrogate ViT model to enhance its robustness (minimizing the inner loss), and then use it to generate perturbations (maximizing the outer loss) that transfer effectively to other models. However, adversarially training a full ViT is computationally expensive due to its large number of parameters.

Figure 5: Study on the effectiveness of diversifying the FFN.

To address this challenge, we introduce a test-time adversarial training strategy by introducing a small number of optimizable tokens to robustify the ViT, thereby improving its effectiveness in transferable attacks with significantly reduced cost.

Table 1: Study on the number of robust tokens. The values in the table represent the average attack success rates of the models.

| # Tokens | 1 | 10 | 20 | 50 | 100 | 200 | 400 |
|---|---|---|---|---|---|---|---|
| MI-FGSM | | | | 54.09 | | | |
| + Dynamic Robust Tokens | 52.89 | 58.45 | 60.09 | 62.68 | 63.29 | 66.68 | 68.33 |
| + Global Robust Tokens | 28.45 | 21.88 | 22.09 | 27.03 | 50.58 | 66.40 | 69.76 |

Concretely, after obtaining the patch embeddings of the input $x$, we append $N_r$ robustification tokens initialized randomly, resulting in the embedding sequence, *i.e.*, $\mathbf{z} = [\mathbf{z}^{\texttt{[CLS]}}, \mathbf{z}^1 + \mathbf{p}_1, \ldots, \mathbf{z}^N + \mathbf{p}_N, \mathbf{z}_r^1, \ldots, \mathbf{z}_r^{N_r}]$, where $\mathbf{z}_r = \{\mathbf{z}_r^i\}_{i=1}^{N_r}$ are the trainable robustification tokens. In each iteration, we first generate an adversarial example of $x$ and update $\mathbf{z}_r$ through adversarial training. Once trained, $\mathbf{z}_r$ is used to enhance the ViT's robustness, and standard attacks are applied to generate transferable adversarial perturbations. The overall objective is:

$$\min_{\mathbf{z}_r} \max_{\delta} \mathcal{L}(f(x + \delta; \mathbf{z}_r), y). \tag{8}$$

However, the above instance-specific online test-time training remains computationally demanding. To further reduce the overhead, we propose an offline strategy that learns a universal set of robustification tokens $\mathbf{z}_r$ on a small calibration dataset $\mathcal{D}$. This offline-learned $\mathbf{z}_r$ can then be appended to any input's token sequence, providing a generic robustness enhancement without per-instance optimization. By precomputing these tokens, we significantly reduce the cost of generating transferable adversarial examples while maintaining strong attack performance.

⚖️ **Results**. As shown in table 1, we conduct experiments to evaluate the impact of robust tokens on adversarial transferability. Both dynamic and global robust tokens can enhance the attack success rates by up to $14\%$. Specifically, using just 10 dynamic robust tokens already leads to an improvement of over $4\%$ in attack success rates. As the number of tokens increases, a consistent improvement is observed, reaching up to $14.24\%$. In contrast, the offline-learned global robust tokens only begin to take effect with 200 tokens, but ultimately achieve better performance, surpassing the dynamic approach by $1.43\%$ with 400 tokens. These results suggest that there exists a trade-off between attack efficiency, including computational overhead, memory consumption, and adversarial transferability, which should be carefully considered in real-world applications.

## 4 Learning to Redundantize for Improved Adversarial Transferability

As aforementioned analysis, the computational redundancy in attention and FFN modules can be amplified by different operations to boost adversarial transferability. To fully leverage these operations to enhance the adversarial transferability, we propose to *learn to redundantize* the ViT on fine-grained transformer blocks. Specifically, we train a stochastic transformation policy that dynamically selects operations to diversify intermediate representations, thereby improving transferability.

We randomly initialize a sampling matrix $\mathbf{M} \in \mathbb{R}^{L \times O}$, where $L$ is the number of transformer blocks and $O$ is the number of possible operations $\phi_o(\cdot)$. Each entry $\mathbf{M}_{l,o}$ denotes the probability of selecting operation $\phi_o$ at the $l$-th block. During each attack iteration, we sample $s < O$ operations per block based on $\mathbf{M}$ and apply the selected operation set $\{\phi_1^l(\cdot), \ldots, \phi_s^l(\cdot)\}$ to the surrogate ViT.

To optimize the distribution $\mathbf{M}$, we treat it as a categorical policy and update it via the REINFORCE estimator. Our objective is to maximize the expected adversarial loss:

$$\max_{\mathbf{M}} \mathbb{E}_{\phi \sim \mathbf{M}} \left[ \mathcal{L}(f(x + \delta(\phi)), y) \right], \tag{9}$$

and the gradient for each entry $\mathbf{M}_{l,o}$ is computed as:

$$\nabla_{\mathbf{M}_{l,o}} \mathcal{L} = -\mathbb{E}_{\phi \sim \mathbf{M}} \left[ \mathcal{L}(f(x + \delta(\phi)), y) \cdot \nabla_{\mathbf{M}_{l,o}} \log P(\phi^l = \phi_o \mid \mathbf{M}_l) \right]. \tag{10}$$

Since $\phi^l$ is drawn from a categorical distribution, we have $\nabla_{\mathbf{M}_{l,o}} \log P(\phi^l = \phi_o) = \frac{1}{\mathbf{M}_{l,o}} \cdot \mathbb{1}[\phi^l = \phi_o]$. Through this gradient-based update, the model learns to emphasize transformations that most improve adversarial transferability in an online and block-specific manner.

Table 2: Our method achieves state-of-the-art performance in attacking diverse models using ViT variants (ViT-B/16, PiT-B, Swin-T) as surrogates. ViT-specific attacks such as TGR, GNS, and FPR are excluded for Swin-T due to unavailable implementations.

| Surrogate | Method | RN-50 | VGG-16 | MN-V2 | Inc-v3 | ViT-B/16 | PiT-B | Vis-S | Swin-T | Avg. |
|---|---|---|---|---|---|---|---|---|---|---|
| ViT-B/16 | MI- | 39.4 | 58.4 | 57.9 | 42.2 | 97.4 | 40.4 | 42.0 | 55.0 | 54.1 |
| | NI- | 40.3 | 59.2 | 58.3 | 44.2 | 96.8 | 41.1 | 44.3 | 57.4 | 55.2 |
| | EMI- | 57.7 | 69.7 | 69.2 | 60.8 | 99.3 | 60.8 | 65.5 | 75.4 | 69.8 |
| | VMI- | 50.3 | 63.6 | 63.2 | 52.7 | 98.3 | 55.7 | 57.4 | 68.1 | 63.7 |
| | PGN | 68.9 | 75.7 | 76.3 | 72.4 | 97.6 | 75.6 | 75.5 | 80.0 | 77.8 |
| | DTA | 43.5 | 65.5 | 64.1 | 48.0 | 99.9 | 46.3 | 49.4 | 62.1 | 59.8 |
| | TGR | 53.4 | 72.5 | 72.4 | 55.5 | 97.7 | 59.2 | 61.8 | 74.5 | 68.4 |
| | GNS | 47.5 | 68.2 | 68.2 | 49.6 | 91.5 | 50.1 | 54.8 | 65.4 | 61.9 |
| | FPR | 52.3 | 66.6 | 68.4 | 52.4 | 97.5 | 56.2 | 60.7 | 71.0 | 65.6 |
| | **Ours** | **77.7** | **90.6** | **91.1** | **79.9** | 99.7 | **78.9** | **83.5** | **93.5** | **86.9** |
| PiT-B | MI- | 39.4 | 58.9 | 56.0 | 38.7 | 26.6 | 95.4 | 44.6 | 48.0 | 44.6 |
| | NI- | 39.7 | 60.4 | 58.4 | 37.3 | 26.0 | 94.2 | 45.8 | 49.4 | 45.3 |
| | EMI- | 58.2 | 71.6 | 72.2 | 57.4 | 46.0 | 98.7 | 66.1 | 69.6 | 63.0 |
| | VMI- | 54.2 | 66.7 | 66.9 | 55.1 | 47.2 | 95.6 | 61.5 | 63.2 | 59.2 |
| | PGN | 71.4 | 77.5 | 78.4 | 73.0 | 69.4 | 93.9 | 77.1 | 79.0 | 75.1 |
| | DTA | 48.6 | 67.8 | 67.5 | 46.4 | 34.7 | **99.9** | 54.9 | 58.4 | 54.0 |
| | TGR | 59.6 | 78.2 | 78.8 | 57.6 | 49.5 | 98.2 | 68.7 | 71.6 | 70.3 |
| | GNS | 58.9 | 78.8 | 77.8 | 58.8 | 46.1 | 98.6 | 68.9 | 71.3 | 69.9 |
| | FPR | 58.3 | 77.5 | 75.1 | 67.8 | 46.1 | 96.4 | 64.4 | 68.6 | 69.3 |
| | **Ours** | **86.0** | **91.7** | **93.6** | **81.0** | **74.1** | 99.7 | **91.7** | **93.4** | **87.4** |
| Swin-T | MI- | 28.8 | 48.1 | 52.8 | 28.8 | 21.3 | 27.0 | 34.1 | 95.7 | 42.1 |
| | NI- | 30.5 | 49.5 | 53.9 | 28.6 | 19.8 | 28.0 | 34.8 | 96.4 | 42.7 |
| | EMI- | 42.2 | 62.4 | 67.8 | 42.3 | 32.4 | 42.9 | 52.2 | **99.7** | 55.2 |
| | VMI- | 49.9 | 61.3 | 68.1 | 48.8 | 46.3 | 54.1 | 60.2 | 97.7 | 60.8 |
| | PGN | 78.5 | 86.8 | 87.8 | 81.8 | 77.7 | 83.4 | 86.9 | 99.3 | 85.3 |
| | DTA | 31.7 | 53.0 | 57.8 | 29.7 | 20.6 | 27.4 | 35.1 | 99.5 | 44.3 |
| | **Ours** | **85.2** | **90.1** | **91.5** | **89.6** | **85.4** | **88.3** | **92.4** | 98.2 | **88.9** |

## 5  Experiments

In our experiment, we fully evaluate the performance of our proposed attacks on different ViTs, including the vanilla ViT, Swin, and PiT. We selected various advanced adversarial attack methods as the baseline to compare, including MI- [Dong et al., 2018], NI- [Lin et al., 2020], EMI- [Wang and He, 2021], VMI-FGSM [Wang et al., 2021b], PGN [Ge et al., 2023], DTA [Yang et al., 2023], TGR [Zhang et al., 2023], GNS [Zhu et al., 2024b], and FPR [Ren et al., 2025b]. For our method, we integrate the learning strategy introduced in section 4 into the MI-FGSM. Following the settings in previous work [Dong et al., 2018, Zhou et al., 2018, Chen et al., 2023], on the ImageNet-1K dataset, we generate $1,000$ adversarial examples by attacking the surrogate model, and evaluate the adversarial transferability by attacking other models.

**ViT as the surrogate model, attack others**. As shown in table 2, our proposed method significantly outperforms all baseline attack methods across all target models, demonstrating superior adversarial transferability. Specifically, our attack achieves an average fooling rate of 86.9%, substantially surpassing the second-best performing method, PGN, which yields an average fooling rate of 77.8%. This highlights the effectiveness of our approach in generating transferable adversarial examples that generalize well across both convolutional and transformer-based architectures.

Notably, the improvement is consistent across a diverse set of models, including both traditional CNNs, *e.g.*, ResNet-50, VGG-16, MobileNetV2, and recent ViT-based architectures *e.g.*, ViT-B/16, PiT-B, Swin-T, and Visformer-S. For instance, on ResNet-50 and VGG-16, our method achieves a fooling rate of 77.7% and 90.6%, respectively, indicating a remarkable gain of over 8 percentage points compared to PGN. Moreover, the attack remains highly effective on vision transformers, achieving near-perfect success rates, *e.g.*, 99.7% on ViT-B/16 and 93.5% on Swin-T, further emphasizing its robustness in the black-box transfer setting.

**PiT as the surrogate model, attack others**. We also evaluate the performance of our proposed attack when using PiT-B as the surrogate model. PiT differs from standard ViT by introducing pooling layers between stages, which reduces computational cost while maintaining competitive performance. Unlike in ViT, where robust tokens are appended to the end of the input sequence, tokens in PiT are arranged in a 2D square matrix. To simplify implementation and preserve the original spatial alignment of the token matrix at its top-left corner, we pad robust tokens only along the right and bottom edges of the matrix. These tokens are then optimized via gradient ascent, following our test-time objective defined in eq. (8).

As shown in table 2, our method again achieves state-of-the-art results across all target models. Compared with PGN, which already performs competitively, our method achieves a further improvement of over 12 percentage points in average fooling rate (87.4% *vs.* 75.1%). This margin is even more pronounced on lightweight convolutional networks, such as MobileNetV2 (93.6% *v.s.* 78.4%) and VGG-16 (91.7% *v.s.* 77.5%), demonstrating the effectiveness of our optimization approach under constrained surrogate architectures. The performance on transformer-based targets remains high as well, with 99.7% on PiT-B and 93.4% on Swin-T, indicating that the learned perturbations are not only strong but also generalizable across different transformer designs.

**Swin as the Surrogate Model, Attacking Others**. Unlike ViT and PiT, the Swin Transformer does not rely on a dedicated classification token. Instead, it generates predictions by aggregating outputs from all tokens in the final transformer stage. Moreover, the fixed input resolution and architectural constraints of Swin Transformer present challenges for integrating a flexible number of robust tokens at the beginning of the input. To overcome this, we adopt a modified strategy by inserting the randomly initialized token embeddings directly into the attention layer and optimizing them via gradient ascent following our test-time objective in eq. (8).

As shown in table 2, our method achieves an average fooling rate of **88.9%**, outperforming all baselines, including PGN (85.3%). It shows strong effectiveness across both convolutional and transformer-based targets, achieving 85.2% on ResNet-50, 90.1% on VGG-16, 85.4% on ViT-B/16, and 98.2% on Swin-T itself. These results underscore the generalizability of our attack strategy, even when launched from a structurally distinct and token-aggregative architecture like Swin.

**Attack Vision-Language Large Models(VLLMs)**. As VLLMs are increasingly adopted in real-world applications, ensuring their robustness is of critical importance. In this work, we evaluate the effectiveness of our proposed adversarial attack method on several widely-used open-source VLLMs, i.e., LLaVA [Liu et al., 2023], Qwen [Bai et al., 2025], InternVL [Chen et al., 2024b] and DeepSeek [Lu et al., 2024]. Specifically, we employ the ViT model as the surrogate model to generate

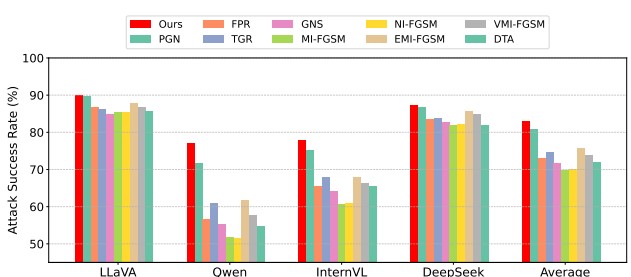

Figure 6: Attack Success Rate on LLaVA-v1.5-7B, Qwen 2.5-VL-7B-Instruct, InternVL 2.5-8B and DeepSeek-VL-7B-Chat. The adversarial examples are generated on ViT-B/16.

adversarial examples. These examples, along with a set of $1,000$ candidate label names, are then input to the VLLMs, which are prompted to select the most appropriate label.

As shown in fig. 6, our method consistently outperforms all baseline approaches, with average improvements of $2.2\%$ against the runner-up method PGN. Notably, on Qwen and InternVL—the two most robust VLLMs in the evaluation—our method surpasses the second-best method by $5.5\%$ and $2.6\%$, respectively. These results highlight that our method consistently generates adversarial examples with high transferability across different VLLMs.

## 6    Conclusion

In this paper, we explore a novel perspective on adversarial attack generation by harnessing the computational redundancy inherent in Vision Transformers (ViTs). Through both theoretical insights and empirical analysis, we demonstrate that data-level and model-level redundancies, traditionally considered inefficient, can be effectively exploited to boost adversarial transferability. We propose a comprehensive framework that integrates multiple redundancy-driven techniques, including attention sparsity manipulation, attention head permutation, clean token regularization, ghost MoE diversification, and test-time adversarial training. Additionally, we introduce an online learning strategy that dynamically adapts redundant operations across transformer layers to further enhance transferability. SOTA performance shown in extensive experiments reveals the overlooked utility of redundancy in ViTs and open new avenues for designing stronger and more transferable adversarial attacks.

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

## A Computational Redundancy in ViTs

We investigate the computational redundancy in Vision Transformers (ViTs) from two complementary perspectives: *data-level* and *model-level* redundancy.

**Data-level redundancy.** We begin by evaluating the robustness of ViTs under partial observations of the input. Specifically, we conduct two types of perturbations: (1) randomly dropping a proportion of patch tokens before entering the transformer, and (2) randomly zeroing out elements in the attention weight matrices during self-attention computation. In both cases, the `[CLS]` token is retained. As shown in Figure 7, the top-1 accuracy on ImageNet remains remarkably stable even after removing up to 50% of tokens or attention weights. This indicates that ViTs possess strong resilience to incomplete or noisy visual evidence, likely due to the high degree of representational redundancy inherent in dense token embeddings and global attention.

**Model-level redundancy.** We further explore the internal redundancy of ViTs by ablating key components of the architecture at inference time. We consider: (1) randomly disabling a subset of attention heads in each layer, and (2) randomly dropping a proportion of hidden units in the intermediate layers of the feedforward network (FFN). As seen in Figure 7, both forms of perturbation lead to graceful degradation in performance. Even with 30–50% of heads or FFN neurons removed, the models still maintain high accuracy. This reinforces the observation that ViTs are significantly overparameterized, and many internal computations can be suppressed without compromising the final output.

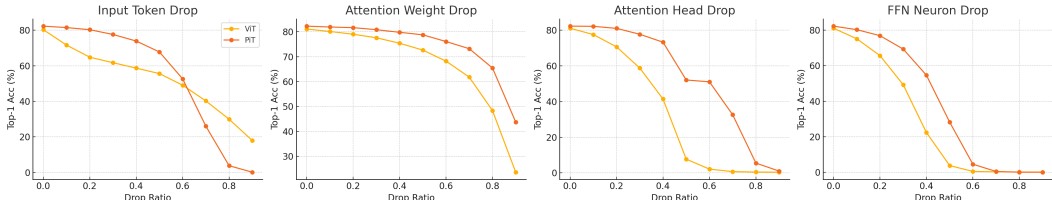

Figure 7: Top-1 accuracy of ViT under various types of token and structural drop perturbations. ViTs exhibit strong robustness to both input-level and architecture-level degradation, suggesting substantial redundancy in both data representation and model computation.

**Related works on studying the computational redundancy of transformers and ViTs.** There have been many works that systematically study and leverage the redundancy within the ViT's architecture. For example, Bolya et al. [2022], Yin et al. [2022], Shang et al. [2024], Arif et al. [2025] find that dropping unimportant visual tokens or merging similar tokens will accelerate the inference of ViTs without harming the model performance. Jin et al. [2024], Fu et al. [2024], He et al. [2024] find that there exists similarity to some degree between different attention heads. Some works leverage the computational redundancy to enhance the performance of model, *e.g.*, the use of MoE [Lin et al., 2024, Chen et al., 2024a].

## B Licenses for existing assets

In our paper, we use the ImageNet as the studied dataset, which is under the BSD 3-Clause License.

