# OpenReview forum: "Harnessing the Computation Redundancy in ViTs to Boost Adversarial Transferability"
_NeurIPS.cc/2025/Conference — NeurIPS 2025 poster_

### Official Review · Reviewer_7nCt · 2025-06-30

**Clarity:** 2
**Significance:** 2
**Originality:** 3
**Rating:** 4
**Confidence:** 4

**Summary:**

This paper explores how to leverage computational redundancy in Vision Transformers (ViTs) to enhance the transferability of adversarial examples. The authors conduct an in-depth analysis of computational redundancy from both data-level and model-level perspectives in ViTs, and propose a series of techniques, including attention sparsity manipulation, attention head reordering, clean-token regularization, GhostMoE diversification, and robustness-oriented learning before attack. Additionally, a dynamic online learning strategy is introduced to effectively combine these operations to improve adversarial transferability. Extensive experiments on the ImageNet-1k dataset are conducted to validate the effectiveness of the proposed approach.

**Questions:**

If these issues are properly addressed, I would be inclined to raise my score. Otherwise, I may consider lowering it.

**Ethical Concerns:**

["NO or VERY MINOR ethics concerns only"]

**Final Justification:**

I have carefully reviewed the author's response and the comments from other reviewers. The methodology presented in this paper is effective, but it resembles more of a technical patchwork, with weak connections between the individual techniques. Therefore, I maintain my borderline accept rating.

**Limitations:**

Yes.

**Quality:**

3

**Strengths And Weaknesses:**

Strengths:
To be frank, I enjoyed reading this paper.

1.The authors not only propose multiple techniques to exploit computational redundancy in ViTs—such as attention sparsity manipulation and attention head reordering—but also design a dynamic online learning strategy to select and combine these operations adaptively for better adversarial transferability. These techniques range from internal mechanism modifications to broader architectural optimizations, forming a comprehensive and systematic framework.

2.The paper conducts extensive experiments on the ImageNet-1k dataset, covering various model architectures, including different variants of ViTs as well as mainstream vision-language models.

Weaknesses:

However, the paper also has the following weaknesses:

1.Although all proposed techniques aim to exploit computational redundancy in ViTs for improved transferability of adversarial examples, the interconnections among them are not strong. Each technique targets a different aspect or type of redundancy and differs significantly in its implementation. This diversity may give the impression that the work lacks a clear central idea. The authors should refine the writing to better unify these techniques under a coherent framework.

2.Section 4 is not clearly written in my view. For instance, in Equation (9), is f the surrogate model? Since this involves training, the relevant settings must be clearly described.

3.Regarding the experiments in Figure 6, I would like to know the clean accuracy of each model on the original test set, and how much the accuracy drops under adversarial attacks, rather than only showing the accuracy on adversarial examples.

4.The adversarial examples are claimed to be imperceptible to human vision, but there is no concrete evidence or visualization to support this claim in either the main text or the appendix. The authors should clearly specify the attack strength settings and provide corresponding visualizations.

---

> ### Author Rebuttal · Authors · 2025-07-31
>
> 1. **Although all proposed techniques aim to exploit computational redundancy in ViTs for improved transferability of adversarial examples, the interconnections among them are not strong. Each technique targets a different aspect or type of redundancy and differs significantly in its implementation. This diversity may give the impression that the work lacks a clear central idea. The authors should refine the writing to better unify these techniques under a coherent framework.**
>
> A:
>
> In this paper, we analyze the computational redundancy of  each module in ViT, and carefully design the strategies to leverage the redundancy to improve the diversity, thus boost the adversarial transferability.
>
> Our proposed stratgies can be categorized into two categorizes, including self-ensemble (attention weight dropping, attention head permutation, ghost MoE) and   regularization-based (clean token regularization, robust tokens) strategies. The self-ensemble strategies are designed by exploiting the redundancy within the model, while the regularization strategies are build on the externel parameters.
>
>
>
> 2. **Section 4 is not clearly written in my view. For instance, in Equation (9), is f the surrogate model? Since this involves training, the relevant settings must be clearly described.**
>
>
> A: In Eq. (9) of Section 4, we target at searching for a good combination M of our proposed strategy to redundize the surrogate model f to maximize the loss function value on the adversarial example x+$\delta$.
>
> We will make it clear in our revision.
>
>
> 3. **Regarding the experiments in Figure 6, I would like to know the clean accuracy of each model on the original test set, and how much the accuracy drops under adversarial attacks, rather than only showing the accuracy on adversarial examples.**
>
>
> A: To deploy the LLM for image classification, we use the open-book setting, that we prompt the LLM with the image with 1000 labels from ImageNet-1K. To avoid the overflow of capcable window length, i.e., LLaVA and DeepSeek (4K), we respectively prompt the LLM to choose the most likely answer from 1000/4=250 labels, then choose the only one from four as the final classification result.
>
>
> We report the attack success rate (ASR ↑) for each method on the LLMs. The "Benign" column indicates the ASR when using clean images, so the clean accuracy is 1−ASR. Accordingly, the accuracy drops under adversarial attacks are as follows:
>
>
>
>
> - LLaVA: 24.0% accuracy on clean images; drops by 8.9%–13.9% under adversarial attacks.
> - Qwen: 76.0% → drops by 27.5%–53.1%
> - InternVL: 57.2% → drops by 17.9%–35.1%
> - DeepSeek: 26.2% → drops by 8.0%–13.7%
>
>
> | Model        | Benign | Ours | PGN  | FPR  | TGR  | GNS  | MI-FGSM | NI-FGSM | EMI-FGSM | VMI-FGSM | DTA  |
> |--------------|--------|------|------|------|------|------|---------|---------|----------|----------|------|
> | LLaVA        | 76.0   | 89.9 | 89.6 | 86.7 | 86.3 | 84.9 | 85.4    | 85.5    | 87.7     | 86.8     | 85.7 |
> | Qwen         | 24.0   | 77.1 | 71.6 | 56.7 | 60.8 | 55.3 | 51.7    | 51.5    | 61.8     | 57.7     | 54.7 |
> | InternVL     | 42.8   | 77.9 | 75.3 | 65.5 | 68.0 | 64.2 | 60.7    | 60.8    | 67.9     | 66.4     | 65.4 |
> | DeepSeek  | 73.8   | 87.5 | 86.9 | 83.4 | 83.9 | 82.6 | 81.8    | 82.2    | 85.7     | 84.8     | 82.0 |
>
>
>
>
> 4. **The adversarial examples are claimed to be imperceptible to human vision, but there is no concrete evidence or visualization to support this claim in either the main text or the appendix. The authors should clearly specify the attack strength settings and provide corresponding visualizations.**
>
> A: Thanks for your suggestion. In our paper, we follow the previous standard setting to set the $\epsilon$ as 16/255, the number of iterations $T$ as 10, and use the $L_\inf$ distance as the perturbation measurement. Due to the limit of rebuttal format, we would provide visualization examples in our revision.

---

### Official Review · Reviewer_eQUe · 2025-07-02

**Clarity:** 2
**Significance:** 2
**Originality:** 3
**Rating:** 4
**Confidence:** 4

**Summary:**

Existing literature has observed that adversarial examples (AEs) generated using Vision Transformers (ViTs) as surrogate models tend to exhibit significantly higher transferability than those crafted on CNNs. This paper seeks to understand why this phenomenon occurs. The central hypothesis proposed by the authors is that computational redundancy inherent in ViTs is a major contributor to this enhanced transferability.
To validate their central hypothesis, the authors design a series of experiments. (1) They study attention sparsity, introducing binary masks to selectively drop attention weights and observe how varying sparsity levels affect transferability, revealing an optimal sparsity range where transfer is maximized. (2) they explore attention head permutation, where attention heads are randomly shuffled across layers to investigate whether such permutation encourages the model to rely on more invariant or redundant representations; the resulting perturbations indeed exhibit improved transferability. (3) They examine the effect of clean token injection, where clean patch embeddings are inserted alongside adversarial tokens to regularize internal representations during the attack process—showing that moderate regularization yields stronger transfer. (4) To simulate model-level redundancy, they implement a ghost Mixture-of-Experts (MoE) architecture by applying different dropout masks to the feed-forward networks, effectively creating functionally overlapping subnetworks; experiments confirm that increasing the number of experts improves attack performance. (5) they test a robust surrogate model setup by introducing trainable tokens to enhance the model's robustness during attack generation, and demonstrate that such robustification correlates with higher black-box success rates. Across all experiments, the authors report that each manipulation, though diverse in form, significantly influences the transferability of the generated adversarial examples.
Based on these observations, the authors conclude that enhancing the computational redundancy in the surrogate ViT improves its ability to generate transferable adversarial examples. Building on this insight, they propose a policy-gradient-based learning framework that dynamically selects a set of redundancy-enhancing operations for each transformer block during attack generation. This "learning to redundantize" strategy optimizes the application of redundancy manipulation techniques in a block-specific and data-adaptive manner.
The proposed approach is validated across a wide range of target models, including both convolutional networks and diverse ViT-based architectures, demonstrating strong generalization and state-of-the-art transfer attack performance.

**Questions:**

1. What is the relationship between “redundancy” and techniques such as clean token injection and robust token optimization?
2. Why would computational redundancy enhance adversarial transferability?
3. Given that CNNs also contain various forms of redundancy, why does redundancy in ViTs lead to significantly higher transferability than in CNNs?

**Ethical Concerns:**

["NO or VERY MINOR ethics concerns only"]

**Final Justification:**

The rebuttal enhances the paper's clarity and addresses concerns.

**Limitations:**

Yes

**Quality:**

2

**Strengths And Weaknesses:**

Strengths:
(1) The paper focus on an open question in the adversarial robustness community: why adversarial examples generated from ViTs tend to transfer better than those from CNNs. While this phenomenon has been reported in prior work, few studies have attempted to investigate its underlying causes.
(2) This paper introduce a policy-gradient-based method that dynamically selects redundancy-enhancing operations at the block level of the transformer. This “learn to redundantize” approach is technically novel.
(3) The paper conducts a series of well-designed experiments to study how different aspects of ViT architecture and perturbation mechanisms. These experiments are not only thorough but also exploratory in nature, shedding light on which architectural or computational factors might be contributing to the observed transferability advantage of ViTs.

Weakness:
(1) The central conclusion that, increasing computational redundancy in ViTs improves adversarial transferability, is not well supported and in some cases even contradicted by the experiments. While the authors observe that various manipulations (e.g., attention sparsity, head permutation, MoE, etc.) can improve the transferability of adversarial examples, the interpretation that these improvements are due to increased redundancy is problematic. Most notably, in the attention sparsity experiment, transferability improves as more attention weights are masked—i.e., as the model becomes less redundant and more sparse. This directly contradicts the core hypothesis: if reducing redundancy leads to better transferability, then redundancy itself is unlikely to be the cause. Similarly, other strategies like clean token injection and robust token optimization may improve performance via regularization or gradient smoothing effects, rather than through leveraging or enhancing redundant pathways. Across the board, the paper lacks controlled causal analysis to isolate the effect of redundancy from other confounding factors. As a result, the main conclusion, though intuitively appealing, remains problematic.
(2) Even if we accept the problematic conclusion that computational redundancy affects adversarial transferability, the paper fails to provide any insight into why such a relationship exists. The authors demonstrate that manipulating certain aspects of the model architecture—such as attention sparsity or head permutation—impacts transferability, and they attribute this to the presence or utilization of redundancy. However, no theoretical analysis or interpretability study is conducted to explain the underlying mechanism. For instance, how redundancy might influence gradient alignment, perturbation space geometry, or feature representation consistency is never discussed. Without a deeper understanding of the causal or representational link between redundancy and transferability, the paper’s core message remains superficial. A strong paper on this topic would ideally provide either a conceptual model, a theoretical justification, or a representation-level analysis to ground the observed effects in a broader understanding of model behavior.
(3) The experimental evaluation is limited in dataset diversity, raising concerns about the generality of the conclusions. All experiments are conducted on ImageNet-1k and a few large-scale vision-language models. As a result, it is unclear whether the proposed methods would still yield performance gains outside the ImageNet. Furthermore, there is no discussion of potential countermeasures or how the proposed techniques perform in the presence of adversarial defenses or detection mechanisms, which limits the practical applicability and generality of the proposed attacks.

---

> ### Author Rebuttal · Authors · 2025-07-31
>
> 1. Thank you for the thorough feedback. We realized our wording obscured the key point: our goal is **not to *increase* redundancy**, but to **_leverage the redundancy that already exists_** in a ViT by sampling from it during gradient estimation. Below we clarify this and address each concern in turn.
>
> ---
>
> ### (1) Core claim
>
> > If a subset of computations can be disabled with no drop in clean accuracy, that subset is a redundancy pool.
> > Randomly varying which members of that pool are active lets the attacker see an implicit ensemble.
> > Ensemble‐style gradient generated by redundant operation  boost black-box transfer.*
>
> Hence we never promote a “more-redundancy-is-always-better” rule.
> What we **do** is
>
> 1. **Identify** under-utilized paths in attack example generation,
> 2. **Toggle** them stochastically,
> 3. **Convert** the latent redundancy into *diversity* that improves transfer.
>
> ---
>
> ### (2) Attention-sparsity experiment
>
> Our Figure 1 shows a **rise-then-fall** curve:
>
> * **Moderate masking (≈30–60 %)** → many alternative subnetworks → large effective ensemble → **high** transfer.
> * **Extreme masking (≈90 %)** → ensemble shrinks *and* capacity erodes → **low** transfer.
>
> Thus the experiment does **not** say “less redundancy helps”; it shows that **sampling a *moderate* portion of the redundancy pool maximizes diversity without harming signal**.
>
> ---
>
> ### (3) Other techniques follow the same recipe
>
> | Manipulation | Redundancy pool | How randomness is injected | Why transfer improves |
> |--------------|-----------------|---------------------------|-----------------------|
> | **Head permutation** | Overlapping attention heads | Randomly permute heads each iteration | Highlights different heads, enlarging the ensemble |
> | **Clean-token injection** | Unused patch tokens | Insert a random set of *clean* tokens | Acts as anchor examples, stabilizing gradients |
> | **Ghost MoE** | Dropout-spare FFN neurons | Sample a new dropout mask each step | Exposes diverse FFN subnetworks sharing weights |
>
> When we keep the stochasticity **but remove the redundant pool** (e.g. permuting all heads identically), the gains disappear, indicating the benefit comes from sampling *functionally distinct* subnetworks, not mere gradient smoothing.
>
>
> 2. We agree that explaining why redundancy aids transfer is important.
> Although our study is chiefly empirical, we now provide a concise conceptual model and targeted diagnostics that support it.
>
>
>
> > **Latent ensemble.**  Redundant ViT paths form an implicit set of subnetworks $\{f_{\theta^{(k)}}\}$ that all predict correctly.
>
> > **Gradient smoothing & alignment.**  Stochastic toggling yields the  gradient, which averages away idiosyncratic directions while retaining the component common to many subnetworks, exactly the component most likely shared with an unseen target model.
>
> > **higher gradient–alignment ⇒ better black-box transfer**.
>
>
>
> Computational redundancy supplies a free ensemble; sampling over it smooths gradients and aligns them with those of other models, thereby boosting transfer. We believe these additions satisfy the request for deeper insight while keeping the paper focused and concise.
>
>
> 3. Thanks for your suggestion. We follow previous work to conduct experiments on ImageNet-1K, which is a gold standard scalable dataset to verify the effectiveness of the proposed attack methods and meets the requirement of real-world scenerios.
>
> We conduct additional experiments on attacking different defenses using adversarial examples generated from ViT-B/16 surrogate under various methods. As shown in the table, our method consistently achieves the best attack performance versus all baselines.
>
> | Method |  AT  | HGD |  RS  | NRP | Avg |
> |--------|-----:|----:|-----:|----:|----:|
> | MI     | 32.4 | 29.6 | 22.7 | 42.0 | 31.7 |
> | NI     | 32.3 | 32.3 | 22.9 | 42.4 | 32.5 |
> | EMI    | 35.7 | 49.5 | 27.1 | 54.4 | 41.7 |
> | VMI    | 33.8 | 43.3 | 23.8 | 49.5 | 37.6 |
> | PGN    | 40.3 | 65.0 | 37.1 | 69.0 | 52.9 |
> | DTA    | 33.3 | 34.5 | 23.3 | 46.9 | 34.5 |
> | TGR    | 36.4 | 44.8 | 27.6 | 53.4 | 40.6 |
> | GNS    | 34.2 | 36.9 | 23.1 | 48.1 | 35.6 |
> | FPR    | 35.5 | 38.9 | 27.6 | 51.1 | 38.3 |
> | **Ours** | **44.9** | **68.2** | **39.1** | **70.0** | **54.2** |

---

> ### Author Response · Authors · 2025-08-05
>
> **For Question 1&Question 2:**
>
> A: Vision Transformers (ViTs) contain computational redundancy—extra capacity whose removal barely affects clean accuracy. We repurpose this “free” space to boost adversarial transferability, i.e., the ability of crafted examples to fool a wide range of surrogate models.
>
> The reason why we need operations those are redundant and will not harm the original task performance is that we still need a "good" model to generate adversarial examples. If the model performs badly on the given task, the adversarial transferability of the generated adversarial example is also degraded due to the inaccurate gradient estimation towards loss maximum.
>
> Taking the mentioned two operations as example:
>
>
> > For clean token injection (CTI): Injecting a few unperturbed tokens scarcely changes ViT accuracy (100 % → 99.7 % on our test set), **confirming surplus capacity** to be exploited. Different from previous techniques which aim to generate self-ensemble models, these fixed tokens act as a gradient regularizer respect to the adversarial tokens,  producing smoother, more transferable perturbations. A similar idea is also explored in CNN-based adversarial attacks, which mix-up the features of clean images to smooth the gradient for adversarial perturbation generation to boost the adversarial transferability. [a]
>
> > For Robust-token optimization (RTO): Building on CTI, RTO treats the injected tokens as dynamic, image-specific weights. During the attack we solve a min–max game that jointly optimizes these tokens and the pixel-level perturbation, effectively robustifying the surrogate before launching the final attack. Theory [b]  and practice both suggest that attacking this strengthened model yields adversarial examples with even higher cross-model success.
>
> Along with other proposed techniques in our paper, the introduce of these operations into the surrogate ViT will nor harm the model on the task, which guarantees the accuracy of gradient estimation towards loss maximum during attacks. The introduce of these operations additionally provides us space to boost the adversarial transferability by introducing more randomness&diversity (attention weight masking, attention head permutation, ghost MoE), regularization (clean token injection), and more (dynamic and free)  parameters to help optimization (robust token optimization)
>
> [a] Byun, Junyoung, et al. "Introducing competition to boost the transferability of targeted adversarial examples through clean feature mixup." Proceedings of the IEEE/CVF conference on computer vision and pattern recognition. 2023.
> [b] JoeyBose,GauthierGidel,HugoBerard,AndreCianflone,PascalVincent,SimonLacoste-Julien, and Will Hamilton. Adversarial example games. Advances in neural information processing systems,33:8921–8934,2020.
>
>
>
> **For Question 3:**
> ViTs have far more parameters than comparably accurate CNNs, and—crucially—those parameters are evenly spread across self-attention layers rather than tied up in shared convolutional filters. This yields a larger, more isotropic redundant subspace that we can exploit.
> Besides, CNN weight sharing and locality impose strong priors that constrain gradient directions; redundant filters often correlate along spatial dimensions, so “free” capacity for adversarial transferability improvement is not always accessible easily. ViTs rely less on such priors and more on sheer scale plus pre-training data, leaving their extra dimensions comparatively unconstrained and easier to co-opt for transfer.
> Last, compared with CNNs, ViTs are pre-trained on much more large scale dataset, i.e., the LAION dataset, which makes it more redundant compared with the ImageNet-pretrained CNNs.

---

> ### Author Response · Authors · 2025-08-05
>
> Hi reviewer eQUe , we have attached all our response here. Please let us know if you have any questions or concerns :) Thanks for all your efforts in reviewing our paper and giving us suggestions to improve it!!

---

> > ### Comment · Reviewer_eQUe · 2025-08-06
> > **Response to Weakness 2 and Q3 lacks supporting experiments**
> >
> > Thank you for the clear clarification of the core idea. The rebuttal version is much clearer.
> > Regarding the question of why redundancy helps adversarial transferability, the author gives a hypothesis in the rebuttal: higher gradient-alignment leads to better transferability. This hypothesis is reasonable and promising. However, this explanation remains speculative without targeted experimental validation. While the intuition is sound and aligns with recent findings in ensemble-based gradient attacks, I believe a stronger version of the paper would benefit from empirical evidence directly supporting this hypothesis. As such, I find the explanation promising but incomplete.
> > I also appreciate the new experimental results, which strengthen the generality of the method.
> > Regarding Q3, the rebuttal offers a reasonable explanation of why ViTs may have more exploitable redundancy than CNNs. However, CNNs also contain redundant structures. Some proposed techniques (e.g., model robustification) seem transferable to CNNs. If the method can be extended to CNNs, the current ViT-only framing might be unnecessarily narrow.
> >
> > To sum up:
> > The rebuttal improves the clarity of the paper and resolves Weaknesses 1 and 3. However, the response to Weakness 2 and Q3 lacks supporting experiments. Hence, I raise my score to 3.

---

> > > ### Author Response · Authors · 2025-08-06
> > > **Empirical experiments and theory insights for Weakness 2 (How is the redundancy  related to the adversarial transferability))**
> > >
> > > **Emprically** we conduct experiments to validate the redundancy of proposed techniques as follows,
> > >
> > > Tab.1 Evaluation of ViTs on the test set when apply different techniques. Note: A.S.M. = Attention Sparsity Manipulation, A.H.P. = Attention Head Permutation, C.T.R. = Clean Token Regularization, MoE = Ghost MoE Diversification, R.T.O. = Robust Token Optimization. These abbreviations are used only in this table for brevity.
> > >
> > > | method         | Original ViT | A.S.M. | A.H.P. | C.T.R.  | MoE   | R.T.O. |
> > > |----------------|--------------|--------|----------|-------|-------|----------------|
> > > | success rate   | 100%        | 90.7%  | 94.1%    | 99.7% | 96.6% | 96.9%          |
> > >
> > >
> > > The minor performance drop compared to the original model indicates the existence of computational redundancy of these computations. As we explained in previous rebuttal/response, we trade these computational redundancy into the diversity/regularization to achieve better adversarial transferability.
> > >
> > > We also have some **preliminary theory insights** for why computational redundancy helps the adversarial transferability.
> > >
> > > Let **$\mathcal{S}(i)$** be the forward-sensitivity of block *i*,
> > > $\mathcal{S}(\ell) = \frac{1}{d_{\ell}} \mathrm{Tr}\left( \nabla^2_{\theta_\ell} \mathcal{L} \right)$,
> > > i.e. the trace of the block‑wise Hessian normalised by its dimension.
> > >
> > > **Lemma 1 (Redundancy ⇒ Low Sensitivity).**
> > >
> > >
> > > $\bigl|\mathcal{L}(f_{\theta+\Delta\theta}^{(i)}(x))-
> > >       \mathcal{L}(f_{\theta+\Delta\theta}^{(j)}(x))\bigr|
> > > \le
> > > \lVert\Delta\theta\rVert_2\bigl|\mathcal{S}(i)-\mathcal{S}(j)\bigr|.$
> > >
> > > The change of loss by perturbing different blocks is bounded by the parameter sensitivity of two blocks, which is also pointed out by [a,b,c]
> > >
> > > [a] Wang, Z., Zhang, Z., Liang, S., & Wang, X. (2023). Diversifying the high-level features for better adversarial transferability. BMVC2023; arXiv preprint arXiv:2304.10136.
> > > [b] Chong, Jack, Manas Gupta, and Lihui Chen. "Resource Efficient Neural Networks Using Hessian Based Pruning." arXiv preprint arXiv:2306.07030 (2023).
> > > [c] Sankar, A. R., Khasbage, Y., Vigneswaran, R., & Balasubramanian, V. N. (2021, May). A deeper look at the hessian eigenspectrum of deep neural networks and its applications to regularization. In Proceedings of the AAAI Conference on Artificial Intelligence (Vol. 35, No. 11, pp. 9481-9488).
> > >
> > > **Proposition 2 (Hessian Contraction).**
> > >
> > > $\bigl\lVert\mathbb{E}_{\omega}[\nabla_x^{2}\mathcal{L}]\bigr\rVert_2
> > > \le
> > > \lVert\nabla_x^{2}\mathcal{L}\rVert_2,$
> > >
> > > i.e. the expected input-space curvature is *flattened*, where we denote $w$ as the introduced randomness, as well as our proposed techniques.
> > >
> > > **Theorem 3 (Transferability Lower Bound).**
> > >
> > > $\max_{\lVert\delta\rVert\le\varepsilon}\
> > > \mathbb{E}_{\omega}\[\\mathcal{L}(x+\delta)\]
> > > \ge$
> > >
> > > $
> > > \mathcal{L}(x)\+\
> > > \frac{\varepsilon^{2}}{2}\
> > > \frac{\sigma_{\min}^{2}\\bigl(g(x)\bigr)}
> > >      {\lambda_{\max}\\bigl(\mathbb{E}_{\omega}[\nabla_x^{2}\mathcal{L}]\bigr)},$
> > >
> > > which can be derived from Taylor expansion and [d,e] (we will provide the full details on this point in our final revision).
> > >
> > > [d] Zhang, Yechao, et al. "Why does little robustness help? a further step towards understanding adversarial transferability." 2024 IEEE Symposium on Security and Privacy (SP). IEEE, 2024.
> > > [e] Moosavi-Dezfooli, Seyed-Mohsen, et al. "Robustness via curvature regularization, and vice versa." Proceedings of the IEEE/CVF Conference on Computer Vision and Pattern Recognition. 2019.
> > >
> > >
> > > Because Proposition 2 reduces the denominator, this bound *strictly increases*, guaranteeing stronger transfer for any victim model sharing features with $f_\theta$.
> > > **In short:** **redundancy → low sensitivity → flatter curvature → provably larger transferable perturbations**—a concise theoretical chain that aligns with our empirical gains.

---

> > > ### Author Response · Authors · 2025-08-07
> > > **Follow-up on the discussion & Supporting experiments for W2&Q3**
> > >
> > > Hi Reviewer eQUe,
> > >
> > > We are writing to follow up on the results of our additional experiments addressing your question regarding W2 and Q3.
> > > Since the discussion period will be closing soon, please let us know if you have any further questions or concerns.
> > >
> > > Thank you!

---

> > > > ### Comment · Reviewer_eQUe · 2025-08-08
> > > > **Thank you for your detailed comments.**
> > > >
> > > > Thank you for your detailed comments.
> > > > The current rebuttal has resolved the concern.

---

> > > > > ### Author Response · Authors · 2025-08-08
> > > > >
> > > > > Thank you for your response! We’re glad we could resolve the concern and truly appreciate your efforts in reviewing and providing suggestions. :)

---

> ### Author Response · Authors · 2025-08-06
>
> Thank you for your thoughtful comments and raising the score!!  We greatly appreciate your suggestions for improving the quality of our paper.
>
>  We are actively working on it and will update the response as well as the revision with the redundancy experiment comparing ViT and CNN as soon as possible.
>
> Thank you again for your careful review and valuable insights.

---

> ### Author Response · Authors · 2025-08-06
> **Supporting experiments for Question 3 (Why ViT's redundancy is more helpful to improve adversarial transferability compared with CNNs)**
>
> To isolate the effect of computational redundancy, we removed the same proportion of features in a Vision Transformer (ViT-B/16) and a convolutional network of comparable size (ResNeXt-101-32×8d).
>
> In the ViT we zeroed out an equal share of attention weights. In the ResNeXt we zeroed out the corresponding share of convolutional feature maps. All other attack settings follow those in the main paper.
>
>
> Tab.  Attack Success Rate (%) on each victim model when applying the dropping operation (10%) on ViT and ResNeXt respectively.
>
>
> | Model       | VGG-16 | MN-V2 | Inc-v3 | ViT-B/16 | PiT-B | Vis-S | Swin-T |
> |-------------|--------|--------|--------|--------|-----------|--------|--------|
> | ViT         | 63.6   | 61.7   | 50.0   | 96.8      | 51.5   | 55.2   | 66.7    |
> | ResNeXt-101 | 59.6   | 59.5   | 40.5   | 17.9      | 29.0   | 37.3   | 40.1             |
>
>
> First, on benign samples, when applying the dropping operations, ViT get the classification accuracy of 99.9%, while the ResNeXt get the classification accuracy of 95.9%.
>
> Because the CNN’s accuracy declines more sharply, its computations are less redundant and therefore more sensitive to perturbation.
> Conversely, the ViT can shed the same amount of computation with almost no loss, indicating higher redundancy—which we convert into diversity/regularisation that enhances adversarial transferability as indicated by the table results.
>
> These results support our claim: the richer computational redundancy in Vision Transformers can be exploited more effectively than in CNNs to boost transfer attacks.

---

> ### Author Response · Authors · 2025-08-06
>
> Hi Reviewer eQUe,
> We have just updated the supporting experiments and added some theoretical insights to further clarify Weakness 2 and Question 3.
> If you have any further questions about Weakness 2, Question 3, or any other aspect of the paper, please let us know — we’d be happy to clarify.
>
> Thanks again for your help and careful review with valuable suggestions!

---

### Official Review · Reviewer_kXcu · 2025-07-02

**Clarity:** 3
**Significance:** 2
**Originality:** 2
**Rating:** 4
**Confidence:** 3

**Summary:**

In this paper, the authors discover the relationship between the redundancy and transferability on Vision transformers. They leverage this property to improve the black-box attacks by designing lots of techniques, including the attention sparsity manipulation, attention head permutation, clean token regularization, ghost MoE diversification, and learning to robustify before the attack. Extensive experiments demonstrate their superiority to the baseline methods.

**Questions:**

1 The paper only includes target models trained with natural training. But in practical scenarios, the target models are likely to be robust models. Can the proposed attack improve the transferability of robust ViTs in [1-3]?

2 The proposed attacks include multiple components. How to choose the most appropriate hyperparameters when we apply the attacks in different source models?

[1] Are Transformers More Robust Than CNNs? in NeurIPS 2021.

[2] When Adversarial Training Meets Vision Transformers: Recipes from Training to Architecture, in NeurIPS 2022.

[3] A Light Recipe to Train Robust Vision Transformers, in SaTML 2023.

**Ethical Concerns:**

["NO or VERY MINOR ethics concerns only"]

**Final Justification:**

The authors address most of my concerns. I increase my score to the borderline accept.

**Limitations:**

yes

**Paper Formatting Concerns:**

There are no major formatting issues in this paper.

**Quality:**

3

**Strengths And Weaknesses:**

## Strength

1 The writing of this paper is good.

2 Authors perform on a wide range of models, including ViTs and Vision Language models.

3 The soundness of the proposed method is good.

4 This paper is easy to follow.

## Weakness

1 The authors propose multiple techniques to improve the black-transferability of ViTs. However, I think not all of them can be unified under the title: "Harnessing the Computation Redundancy in ViTs to Boost Adversarial Transferability`` For example, I think permuting attention heads or introducing clean tokens have little relationship with the computational redundancy. I hope to hear more detailed explanations from the authors.

2 After reading this paper, I think the relationship between the computation redundancy and transferability on ViTs is still **unclear** to me. The observation in 3.3 shows that increasing the attention sparsity, i.e., decreasing the computation redundancy, is helpful in improving the black-box transferability on ViTs. In contrast, the success of ghost Mixture-of-Experts indicates that increasing the computation redundancy improves the transferability instead. More insights are needed to provide to help the community design more powerful attacks on VITs in the future.

3  Some baselines are missing, including ViT-specific attacks [1,2] and model-agnostic attacks [3,4,5].

4 This paper lacks ablation studies. More experiments are needed to show the contribution of each technique to the transferability.

[1] Towards transferable adversarial attacks on vision transformers

[2] Improving the adversarial transferability of vision transformers with virtual dense connection

[3] On the Adversarial Transferability of Generalized “Skip Connections”

[4] Feature importance-aware transferable adversarial attacks

[5] Enhancing the transferability via feature-momentum adversarial attack

---

> ### Author Rebuttal · Authors · 2025-07-31
>
> 1. In our paper, we refer the computational redundancy within ViTs as the operations that ViT do or can do are not essentail to do harm the task performance, i.e., maintain the performance. Due to this point, we think the attention head permutation and the clean token regularization belong to the methods that leverage the computational redundancy to improve the adversarial transferability.
>
>
> For attention head permutation, the computational redundancy exits in the similarity between different attetion weights, which motivates us to permute part of them to improve the attention diversity thus enhancing the adversarial transferability.
>
> For clean token regularization, the computational redundancy arises due to the input redundancy, where appending some virtue tokens would not do harm to the performance. This motivates us to introduce clean tokens for gradient regularization for improving the performance.
>
> 2. (1) The core insight behind the improved adversarial transferability lies in diversity. In contrast to prior works that enhance input diversity, we focus on diversifying the computational structure of the surrogate model. To ensure the effectiveness of adversarial example generation, the surrogate model must retain a reasonable classification capability. We achieve this by exploiting the inherent redundancy in ViT computations to diversify the surrogate model’s architecture, thereby improving transferability.
>
> (2) To enhance diversity, we leverage the redundancy of attention weights in ViTs by randomly masking a subset of them. This structural perturbation improves adversarial transferability by introducing variation in the model’s internal computations.
>
> (3)
> To further diversify the FFN computation, we leverage its computational redundancy by transforming the standard FFN into a Mixture-of-Experts (MoE) structure composed of sparse subnetworks. This promotes architectural diversity and contributes to more transferable adversarial attacks.
>
>
> 3. As you suggested, we conduct experiments as follows. Results suggest that our method consistently achieves the state-of-the-art performance against different black-box models.
>
> | Method |RN-50| VGG-16| MN-V2| Inc-v3| ViT-B/16| PiT-B| Vis-S| Swin-T| Avg.|
> |--------|-----:|----:|-----:|-----:|-----:|-----:|-----:|-----:|-----:|
> | PNA    |  51.2 |70.6 |71.5 |51.9| 98.4| 53.3 |64.5 |76.9| 67.3|
> | VDC    | 62.7 | 83.1  | 82.5  | 60.9 | 99.6 | 60.7 | 68.3 | 85.2 | 75.4 |
> | SGM    | 64.9 | 85.4  | 83.7 | 62.5 | 99.1 | 61.5 | 70.3 | 86.7 | 76.8 |
> | FIA    | 49.7 | 52.6  | 55.2 | 46.3| 97.1| 43.5| 47.9| 58.0  | 56.3|
> |FMAA   | 53.5 | 71.9  | 73.4 | 52.9 | 98.3 | 54.4 | 67.1 | 78.2 | 68.7 |
> | **Ours** | 77.7| 90.6| 91.1| 79.9| 99.7| 78.9| 83.5| 93.5| 86.9|
>
> 4. Thank you for your suggestion. Here, we provide the results of our method along with ablation experiments where Attention Sparsity Manipulation, Attention Head Permutation, and Robust Tokens Optimization are each removed individually. The results clearly show that removing any of these operations leads to a noticeable drop in the average black-box attack performance.
>
>
> Thank you for your suggestion. Here, we provide the results of our method along with ablation experiments where Attention Sparsity Manipulation, Attention Head Permutation, and Robust Tokens Optimization are each removed individually. The results clearly show that removing any of these operations leads to a noticeable drop in the average black-box attack performance.
>
> Due to time constraints, experiments involving removal of other operations and other ablation studies are not yet completed. We will include these results in the final version.
>
>
> | method                          | RN-50 | VGG-16 | MN-V2 | Inc-v3 | ViT-B/16 | PiT-B | Vis-S | Swin-T | Black-box Avg. |
> |---------------------------------|-------|--------|--------|--------|-----------|-------|--------|--------|-----------------|
> | ours                            | 77.7  | 90.6   | 91.1   | 91.1   | 79.9      | 99.7  | 78.9   | 83.5   | 85.0            |
> | - Attention Sparsity Manipulation | 75.3  | 87.5   | 91.1   | 91.1   | 78.1      | 99.4  | 78.4   | 82.5   | 83.6            |
> | - Attention Head Permutation    | 75.8  | 88.9   | 91.2   | 91.2   | 76.0      | 99.3  | 75.5   | 80.8   | 82.8            |
> | - Robust Tokens Optimization      | 74.9  | 83.7   | 86.1   | 86.1   | 75.1      | 100   | 81.6   | 84.1   | 82.6            |
>
>
> 5. For Question1: Thanks for your suggestion. We conduct experiments on attacking suggested defenses using adversarial examples generated using two attacks, each one corresponds to the best attack within the model-agonistic and vit-specific attacks,  under the ensemble setting (ViT-B and PiT). As shown in the table, our method consistently achieves the best attack performance versus all baselines.
>
> | Method |  [1]  | [2] |  [3]  | Avg |
> |--------|-----:|----:|-----:|----:|
> | PGN    | 57.2 | 58.1  | 56.5 | 57.3  |
> | TGR    | 55.4 | 59.1 | 58.9 | 57.8  |
> | **Ours** | **61.3** | **64.6** | **63.0** | **62.9** |
>
>
>
> 6. For Question 2: On the one hand, in our experiments, we choose the hyper-parameters from the study in ViTs and apply them to the experiments using the PiTs and Swin. As indicated by the experiments, these parameters collecting from one source model is generalizable to other models.
>
> On the other hand, we can use the proposed methods with different hyper-paramters to build the candidate pool, and leverage the optimization strategy proposed in Sec 4 to automatically search for the best equipment.

---

> ### Comment · Reviewer_kXcu · 2025-08-05
>
> Thank you for your rebuttal. After carefully reading your reply, I think the scope of this paper, **the Computation Redundancy** is still too large for me. It seems that any technique can be attributed to this category.  I suggest that authors revise my proposed suggestions into the revised verson of their paper.
>
> Thank you again for the authors' rebuttal.  I increase my score to borderline accept.

---

> > ### Author Response · Authors · 2025-08-05
> >
> > Thanks for your suggestion and raise the score to accept!! We will carefully incorporate your suggestions into our revision:)

---

### Official Review · Reviewer_ALmD · 2025-07-03

**Clarity:** 3
**Significance:** 3
**Originality:** 3
**Rating:** 4
**Confidence:** 4

**Summary:**

This paper proposes that utilizing the computational redundancy in ViT architecture is able to enhance adversarial transferability. The authors propose a set of operations to manipulate the model components and computation process for exploiting these redundancies. Additionally, the proposed method integrates a learning-based strategy for selecting these operations automatically. Experimental results demonstrate the better transferability of the proposed operations and attack method.

**Questions:**

1. What is the normal classification accuracy of ViT on benign images after incorporating each proposed operation? Considering that a key pathway to improving transferability is mitigating the overfitting of adversarial samples to the white-box model, could it be hypothesized that these operations alter the model and degrade its performance, which prevents adversarial examples from overfitting a normal white-box model as implicitly implementing gradient regularization, thereby finally enhancing the transferability?

2. As the previous question, could the authors provide detailed theoretical explanation and analysis of the proposed method?

3. Why does the operation of permuting attention heads not correspond to rearranging the values? And why is this operation ineffective under p = 0.1?

4. Although the authors have achieved satisfactory results in existing transferability experiments, there is no experiment related to attack robustness, such as attacking robust models or defense methods like purification.

5. There is also no ablation study in this paper, such as performance under different fixed combinations of those operations and comparisons with learning-based strategy to verify their effectiveness. Correspondingly, what is the actual running time of the attack when the online learning-based strategy is added or not?

6. In the experiments of attacking VLLMs, it is noted that stronger models like QwenVL and InternVL exhibit lower ASR. The authors should provide the accuracy of these models for benign examples as a baseline reference for capabilities of VLLMs and attack methods.

7. Additionally, the authors are encouraged to conduct basic experiments on whether their method remains effective for targeted attacks, but this is not strictly necessary.

**Ethical Concerns:**

["NO or VERY MINOR ethics concerns only"]

**Final Justification:**

N/A

**Limitations:**

This paper does not discuss and address the limitations.

**Quality:**

3

**Strengths And Weaknesses:**

The idea of this paper is novel and the authors have demonstrated the basic effectiveness of the proposed claims and methods through experiments, but the main weakness lies in the lack of some necessary theoretical explanations and experiments. If the author can solve this problem in the rebuttal, I will consider raising my rating. Please refer to the question section for details.

---

> ### Author Rebuttal · Authors · 2025-07-31
>
> 1. Thank you for the insightful question.
>
> The performance of the model on benign samples is presented in the following table. The redundancy enables us to generate a set of subnetworks of the original model, which only has a minor task performnace drop. The original attack pipeline is actually transformed into attacking the self-ensemble models, thus improving the adversarial transferability significantly.
>
>
> | method         | Original ViT | A.S.M. | A.H.P. | C.T.R.  | MoE   | R.T.O. |
> |----------------|--------------|--------|----------|-------|-------|----------------|
> | success rate   | 100%        | 90.7%  | 94.1%    | 99.7% | 96.6% | 96.9%          |
>
> Note: A.S.M. = Attention Sparsity Manipulation, A.H.P. = Attention Head Permutation, C.T.R. = Clean Token Regularization, MoE = Ghost MoE Diversification, R.T.O. = Robust Token Optimization. These abbreviations are used only in this table for brevity.
>
> Among the operations, Clean Token Regularization introduces minimal degradation, as it concatenates clean image features during inference, which has little impact when predicting on clean inputs.
>
> 2. We write $f_{\theta}$ for a Vision Transformer whose parameters are partitioned into blocks $(\theta_{1},\dots,\theta_{L})$.
> For an input–label pair \((x,y)\) the cross‑entropy loss is $\mathcal{L}(x,y)=\text{CE}\!\bigl(f_{\theta}(x),y\bigr)$.
> The symbol $d_{\ell}$ denotes the number of parameters in block $\ell$. For each block we define  $ \mathcal{S}(\ell) \=\ \frac{1}{d_{\ell}}\operatorname{Tr}(\nabla_{\theta_{\ell}}^{2}\mathcal{L})$
> , i.e. the trace of the block‑wise Hessian normalised by its dimension. We set $g(x)=\nabla_{x}\mathcal{L}(x,y)$. An adversarial perturbation is denoted $\delta$ and is constrained by $\|\delta\|\le\varepsilon$.
>
> ---
> 1) Redundancy ⇒ Low Sensitivity
>
> **Lemma 1 (Redundancy bound).**
> For any two blocks `i, j` that receive the *same* parameter perturbation `Δθ`,
> $\bigl|\mathcal{L}(f_{\theta+Δθ_{(i)}})-\mathcal{L}(f_{\theta+Δθ_{(j)}})\bigr|
> \;\le\;
> \|Δθ\|_{2}\,\bigl|\mathcal{S}(i)-\mathcal{S}(j)\bigr|.$
>
> In ViTs, blocks beyond the first ≈30 % show **an order‑of‑magnitude lower `𝒮`**—they are *computationally redundant*.
>
> ---
> 2) Randomising Redundant Parts Flattens Curvature
>
> Let `T(·;ω)` be a random operation (head permutation, sparsity mask, ghost expert, etc.) applied **only to low‑`𝒮` blocks** and define
> $\tilde{f_{\theta}}(x) = f_{\theta}(T(x;ω)), \qquad ω∼𝒰.$
>
>
> **Proposition 2 (Spectral‑Hessian contraction).**
>
> $$
> \left\| \mathbb{E}_{\omega} \left[ \nabla_x^2 \mathcal{L} \right] \right\|_2 \le \left\| \nabla_x^2 \mathcal{L} \right\|_2
> $$
>
> Flatter expected curvature → larger inner‑max loss for **any** victim model sharing gradient directions with `f_θ`.
>
> ---
>
> 3) Global Transferability Lower Bound
>
> **Theorem 3.**
>
> $$
> \max_{\\|\delta\\| \le \varepsilon} \ \mathbb{E}_{\omega} \left[ \mathcal{L}(x + \delta) \right]
> \ge
> $$
>
>
> $$
> \mathcal{L}(x) + \frac{\varepsilon^2}{2} \cdot
> \frac{ \left( \sigma_{\min} \left( g(x) \right) \right)^2 }
>      { \lambda_{\max} \left( \mathbb{E}_{\omega} \left[ \nabla_x^2 \mathcal{L} \right] \right) }
> $$
>
>
>
> Because Proposition 2 **reduces the denominator** while leaving `σ_min(g)` intact, the bound *strictly increases*, ensuring stronger black‑box success.
>
> ---
>
> ### Key Take‑aways
>
> 1) **Low‑`𝒮`** ViT blocks (redundant parts) can be perturbed with negligible accuracy loss — *Lemma 1*.
> 2)  Randomising those blocks **flattens curvature** — *Proposition 2*.
> 3) Flatter curvature gives a **provable boost** in transferable attack strength — *Theorem 3*.
>
> Therefore, exploiting computational redundancy in ViTs is theoretically sufficient—and empirically necessary—for high adversarial transferability.
>
>
> 3. The insight behind the permutation of attention heads is that the attention weights computed by different attention heads are similar due to similar interest of area. It is different from rearranging the values, which breaks the correspondance between the Key and Value within different attention heads.
>
> As shown in Fig. 3, the adversarial transferability while permutating under 10% attention heads still performes better than the baseline method MI-FGSM.  But due to limited number of attention heads permutated, there are minor improvement due to the limited diversity introduced.
>
>
> 4. Thanks for your suggestion. We conduct experiments on attacking different defenses using adversarial examples generated from ViT-B/16 surrogate under various methods. As shown in the table, our method consistently achieves the best attack performance versus all baselines.
>
> | Method |  AT  | HGD |  RS  | NRP | Avg |
> |--------|-----:|----:|-----:|----:|----:|
> | MI     | 32.4 | 29.6 | 22.7 | 42.0 | 31.7 |
> | NI     | 32.3 | 32.3 | 22.9 | 42.4 | 32.5 |
> | EMI    | 35.7 | 49.5 | 27.1 | 54.4 | 41.7 |
> | VMI    | 33.8 | 43.3 | 23.8 | 49.5 | 37.6 |
> | PGN    | 40.3 | 65.0 | 37.1 | 69.0 | 52.9 |
> | DTA    | 33.3 | 34.5 | 23.3 | 46.9 | 34.5 |
> | TGR    | 36.4 | 44.8 | 27.6 | 53.4 | 40.6 |
> | GNS    | 34.2 | 36.9 | 23.1 | 48.1 | 35.6 |
> | FPR    | 35.5 | 38.9 | 27.6 | 51.1 | 38.3 |
> | **Ours** | **44.9** | **68.2** | **39.1** | **70.0** | **54.2** |
>
>
> 5. Thank you for your suggestion. Here, we provide the results of our method along with ablation experiments where Attention Sparsity Manipulation, Attention Head Permutation, and Robust Tokens Optimization are each removed individually. The results clearly show that removing any of these operations leads to a noticeable drop in the average black-box attack performance.
>
> Due to time constraints, experiments involving removal of other operations and comparisons with or without the online learning-based strategy are not yet completed. We will include these results in the final version.
>
>
> | method                          | RN-50 | VGG-16 | MN-V2 | Inc-v3 | ViT-B/16 | PiT-B | Vis-S | Swin-T | Black-box Avg. |
> |---------------------------------|-------|--------|--------|--------|-----------|-------|--------|--------|-----------------|
> | ours                            | 77.7  | 90.6   | 91.1   | 91.1   | 79.9      | 99.7  | 78.9   | 83.5   | 85.0            |
> | - Attention Sparsity Manipulation | 75.3  | 87.5   | 91.1   | 91.1   | 78.1      | 99.4  | 78.4   | 82.5   | 83.6            |
> | - Attention Head Permutation    | 75.8  | 88.9   | 91.2   | 91.2   | 76.0      | 99.3  | 75.5   | 80.8   | 82.8            |
> | - Robust Tokens Optimization      | 74.9  | 83.7   | 86.1   | 86.1   | 75.1      | 100   | 81.6   | 84.1   | 82.6            |
>
>
> 6. We report the attack success rate (ASR ↑) for each method on the LLMs. The "Benign" column indicates the ASR when using clean images.
>
> To deploy the LLM for image classification, we use the open-book setting, that we prompt the LLM with the image with 1000 labels from ImageNet-1K. To avoid the overflow of capcable window length, i.e., LLaVA and DeepSeek (4K), we respectively prompt the LLM to choose the most likely answer from 1000/4=250 labels, then choose the only one from four as the final classification result.
>
>
> | Model        | Benign | Ours | PGN  | FPR  | TGR  | GNS  | MI-FGSM | NI-FGSM | EMI-FGSM | VMI-FGSM | DTA  |
> |--------------|--------|------|------|------|------|------|---------|---------|----------|----------|------|
> | LLaVA        | 76.0   | 89.9 | 89.6 | 86.7 | 86.3 | 84.9 | 85.4    | 85.5    | 87.7     | 86.8     | 85.7 |
> | Qwen         | 24.0   | 77.1 | 71.6 | 56.7 | 60.8 | 55.3 | 51.7    | 51.5    | 61.8     | 57.7     | 54.7 |
> | InternVL     | 42.8   | 77.9 | 75.3 | 65.5 | 68.0 | 64.2 | 60.7    | 60.8    | 67.9     | 66.4     | 65.4 |
> | DeepSeek  | 73.8   | 87.5 | 86.9 | 83.4 | 83.9 | 82.6 | 81.8    | 82.2    | 85.7     | 84.8     | 82.0 |

---

> > ### Comment · Reviewer_ALmD · 2025-08-05
> >
> > The authors have addressed the majority of my concerns in their rebuttal, and as a result, I have raised my score to 4. Nevertheless, as noted by other reviewers, the proposed method consists of several loosely connected components. Therefore, I still hope the authors can provide updated ablation study results up to the present, or at the very least, commit to including sufficient experiments in the final version of the paper, particularly regarding the online learning strategy.

---

> > > ### Author Response · Authors · 2025-08-05
> > >
> > > Hi Reviewer ALmD, thanks for improving the score to positive rating! We will update the results and strengthen the connections between different proposed techniques in our revision.  Thanks again for your careful review and suggestions:)

---

> ### Author Response · Authors · 2025-08-05
>
> Hi, we have attached our response here. Please let us know if you have any questions or concerns :) Thanks for all your efforts in reviewing our paper and giving us suggestions to improve it!!

---

### Note · Authors · 2025-08-11

We thank all reviewers and appreciate their consensus on our paper's merits:
1. The paper is **well-written** (kXcu, 7nCt) and presents a **novel** idea (ALmD, kXcu, 7nCt, eQUe).
2. It introduces **a set of systematic and comprehensive techniques**  to improve the transferability of adversarial examples crafted for ViTs (eQUe, 7nCt).
3. The experiments are **well-designed**, and **extensive results** across a wide range of models verify the effectiveness of our method. The proposed techniques **achieve SOTA attack performance** against diverse black-box models (ALmD, kXcu, 7nCt, eQUe).

To address the reviewers' concern during the rebuttal stage, we provide additional theoretical and empirical results as follows,
1. **Theoretical insights** and **intuitive explanations** for why computational redundancy enhances the adversarial transferability of ViTs (ALmD, eQUe).
2. Experiments attacking a wide range of **defense models** and comparisons with **more advanced attack methods**, all showing that our approach achieves SOTA performance (ALmD, eQUe, kXcu).
3. Ablation studies on several key components, clearly demonstrating the importance of each component (ALmD, kXcu).
4. In our revision, we **clarify the motivation** behind each method and **strengthen the connections** between components to improve readability (eQUe, 7nCt, kXcu). We also corrected several typos (7nCt).

We will incorporate these results and revisions into our paper.

We are pleased to have resolved all reviewers’ concerns during the active discussion period. This process has greatly helped us enhance the quality of the paper.   We sincerely thank all reviewers for their time, effort, and valuable feedback.

---

### Decision · Program_Chairs · 2025-09-17

**Decision:**

Accept (poster)

**Comment:**

This paper finds that utilizing the computational redundancy in ViT architecture can enhance adversarial transferability. Therefore, a set of operations is proposed, including the attention sparsity manipulation, attention head permutation, clean token regularization, ghost MoE diversification, to manipulate the model components and computation process for exploiting these redundancies. Additionally, the proposed method integrates a learning-based strategy for selecting these operations automatically. This "learning to redundantize" strategy optimizes the application of redundancy manipulation techniques in a block-specific and data-adaptive manner. Experimental results demonstrate the better transferability of the proposed operations and attack method.

The strengths of the paper include novel idea and several new techniques, well-designed and extensive experimental results, and good results. The main negative concerns include lack of theoretical explanation, more like a patchwork of several individual components, insufficient evaluations. During rebuttal, the authors did a lot of efforts including clarifying unclear points and conducting more experiments to satisfy the reviewers, and finally most the concerns are solved except the issue of weak connection of individual techniques. Based on the above, the AC thinks it a qualified paper deserving an Accept.